# Healthy-like CD4^+^ Regulatory and CD4^+^ Conventional T-Cell Receptor Repertoires Predict Protection from GVHD Following Donor Lymphocyte Infusion

**DOI:** 10.3390/ijms231810914

**Published:** 2022-09-18

**Authors:** Jessica Schneider, Leonie Kuhlmann, Yankai Xiao, Solaiman Raha, Günter Bernhardt, Michael Stadler, Felicitas Thol, Michael Heuser, Matthias Eder, Arnold Ganser, Sarina Ravens, Reinhold Förster, Immo Prinz, Christian Koenecke, Christian R. Schultze-Florey

**Affiliations:** 1Department of Hematology, Hemostasis, Oncology and Stem Cell Transplantation, Hannover Medical School, 30625 Hannover, Germany; 2Institute of Immunology, Hannover Medical School, 30625 Hannover, Germany; 3Institute of Systems Immunology, University Medical Center Hamburg-Eppendorf, 20251 Hamburg, Germany

**Keywords:** donor lymphocyte infusion, immunotherapy, graft-versus-host disease, allogeneic hematopoietic stem-cell transplantation, T_reg_ cells, *TRB* sequencing, TCR repertoire

## Abstract

Donor lymphocyte infusion (DLI) can (re-)induce durable remission in relapsing patients after allogeneic hematopoietic stem-cell transplantation (alloHSCT). However, DLI harbors the risk of increased non-relapse mortality due to the co-occurrence of graft-versus-host disease (GVHD). GVHD onset may be caused or accompanied by changes in the clonal T-cell receptor (TCR) repertoire. To investigate this, we analyzed T cells in a cohort of 21 patients receiving DLI after alloHSCT. We performed deep T-cell receptor β (*TRB*) sequencing of sorted CD4^+^CD25^+^CD127^low^ regulatory T cells (T_reg_ cells) and CD4^+^ conventional T cells (T_con_ cells) in order to track longitudinal changes in the TCR repertoire. GVHD following DLI was associated with less diverse but clonally expanded CD4^+^CD25^+^CD127^low^ T_reg_ and CD4^+^ T_con_ TCR repertoires, while patients without GVHD exhibited healthy-like repertoire properties. Moreover, the diversification of the repertoires upon GVHD treatment was linked to steroid-sensitive GVHD, whereas decreased diversity was observed in steroid-refractory GVHD. Finally, the unbiased sample analysis revealed that the healthy-like attributes of the CD4^+^CD25^+^CD127^low^ T_reg_ TCR repertoire were associated with reduced GVHD incidence. In conclusion, CD4^+^CD25^+^CD127^low^ T_reg_ and CD4^+^ T_con_ *TRB* repertoire dynamics may provide a helpful real-time tool to improve the diagnosis and monitoring of treatment in GVHD following DLI.

## 1. Introduction

AML relapse is the main cause of mortality after allogeneic hematopoietic stem-cell transplantation (alloHSCT). Donor lymphocyte infusions (DLIs) are used as a curative treatment option in relapsing patients (therapeutic DLIs). They are also applied for the conversion of mixed chimerism or persisting minimal residual disease (pre-emptive DLIs) as well as to prevent relapse in high-risk disease (prophylactic DLIs) [1]. The donor T cells are transferred to (re-)induce a graft versus leukemia (GVL) reaction eliminating the recurring or persisting malignant cells [2,3]. However, in contrast with specific cell therapies directly targeting the malignant cells, such as CAR T-cell therapy, DLI is an unselective T-cell approach relying on allo-reactivity of the donor T cells against leukemia antigens. Therefore, the induction of graft-versus-host disease (GVHD) may occur, either simultaneously or even exclusively. Indeed, GVHD conveys the major toxicity in the setting of DLI and is the main driver of non-relapse mortality [4]. Elevated amounts of CD3^+^ cells within the DLI product increase the risk of GVHD induction; therefore, repeated DLIs using a dose-escalating regimen have been introduced with the goal to separate the induction of GVL from GVHD [5]. However, while the GVL response to DLI can be measured using a high-sensitivity chimerism analysis [6], the onset of GVHD cannot be predicted prior to the clinical occurrence, as suitable biomarkers are largely missing. From DLI treatment trials in CML, it emerged that GVL and GVHD can occur separately, indicating the dependence on leukemia- and/or host-antigen-specific T-cell clones [7]. Thus, an in-depth analysis of the DLI immune-cell repertoire might allow the linking of immune-cell repertoire dynamics with allo-reactive treatment response to be performed.

A recent report analyzing the cell subset composition of DLI products suggested that a higher proportion of CD27^+^ B cells, as well as naïve CD8^+^ cells, CD27^+^ NK cells and mononuclear cells, is associated with higher rates of GVHD [8]. Surprisingly, a higher fraction of T regulatory (T_reg_) cells was not protective for GVHD in that study. T_reg_ cells have been described as the main T-cell subset maintaining self-tolerance in the immune system. Consequently, they contribute to the prevention of autoimmune and inflammatory diseases such as GVHD [9]. The adoptive transfer of T_reg_ cells was shown to prevent GVHD while retaining GVL activity after alloHSCT [10,11], and the application of T_reg_-depleted DLI was shown to improve GVL [12].

The use of the T-cell receptor (TCR) repertoire analysis has improved our knowledge about overall T-cell diversity kinetics, including the outgrowth of individual specific T-cell clones. Easy-to-handle tools to decipher TCR-based repertoire dynamics in the context of DLI are just evolving and mostly involve spectratyping [13,14,15]. However, this technique only allows a rough estimate of repertoire diversity to be conducted, and the frequency of individual TCRs cannot be detected. This can be overcome by applying deep sequencing technology, as a direct measure of high-resolution TCR diversity, thereby enabling the observation of the underlying immune-cell mechanism in alloHSCT patients at the clonal level [16,17,18,19]. With regard to GVHD, we previously reported that an expanded T_reg_ TCR repertoire in the peripheral blood early after alloHSCT was associated with protection from GVHD within the first 100 days post alloHSCT [20]. In an unsorted, bulk CD3^+^ TCR repertoire analysis after alloHSCT, reduced diversity and clonal expansion were observed in acute GVHD (aGVHD) [21,22,23,24]. Moreover, treatment-refractory gastrointestinal aGVHD was linked to a clonally expanded CD3^+^ TCR repertoire in the blood [25].

In the setting of DLI, where immune-cell and TCR reconstitution has fully taken place, the role of immune-cell repertoire dynamics with regard to the risk of developing GVHD remains to be elucidated. Here, in a cohort of patients receiving DLI after alloHSCT, we longitudinally assessed the TCR repertoire via the deep sequencing of the TCR beta chains of sorted CD4^+^CD25^+^CD127^low^ T_reg_ cells and sorted CD4^+^ T_con_ cells while observing the emergence and clinical course of GVHD during a study period of 36 months post DLI. In this study, we show that patients without GVHD post DLI exhibited broad CD4^+^CD25^+^CD127^low^ T_reg_ and CD4^+^ T_con_ *TRB* repertoires similar to healthy individuals. Moreover, we observed a diversification of the CD4^+^CD25^+^CD127^low^ T_reg_ and CD4^+^ T_con_ repertoires upon steroid treatment, which was indicative of GVHD treatment response. Finally, we examined whether the features of healthy-like repertoires early after DLI could predict the occurrence of GVHD. Overall, these findings could guide the diagnosis and monitoring of treatment in GVHD following DLI.

## 2. Results

### 2.1. Cohort Analysis

A total of 29 patients were recruited to this prospective observational study, of whom 21 met the inclusion criteria. Nine healthy subjects served as the control group (Figure 1). Patient characteristics and details on DLI treatment are given in Appendix A. Healthy controls’ characteristics are shown in Appendix A. GVHD was thoroughly monitored during the study period of 36 months post DLI. GVHD was diagnosed in 12/21 patients, among whom 9 patients required treatment with systemic steroids. In 4/9 cases, additional immunosuppression was necessary (Figure 2). Details on GVHD characteristics and treatment are shown in Appendix A. The comparison of the GVHD and noGVHD groups revealed a higher number of DLIs and thus a higher number of applied donor T cells in patients with GVHD (Table 1). Relapse occurred 8 months earlier in patients without GVHD compared with GVHD patients (3.7 vs. 11.7 months post DLI; *p* = 0.057; Table 1). This was confirmed via a competing risk analysis showing a significantly lower cumulative incidence of relapse (CIR; *p* = 0.024) and a trend towards higher NRM (*p* = 0.067) for GVHD compared with noGVHD during the 36 months of follow-up (Appendix A).

### 2.2. Patients without GVHD Exhibited a Healthy-LIKE CD4^+^CD25^+^CD127^low^ T_reg_ and CD4^+^ T_con_ Repertoire Diversity

To assess clonal changes in the CD4^+^CD25^+^CD127^low^ T_reg_ and CD4^+^ T_con_ *TRB* repertoires at the occurrence of GVHD after DLI, we sequenced the CDR3 region of the *TRB* chains from sorted CD4^+^CD25^+^CD127^low^ T_reg_ and CD4^+^ T_con_ cells. To test whether diverging cell counts of patient and HC samples influenced the repertoire diversity, evaluated via the inverse Simpson index (1/D), we performed correlational analyses between absolute cell counts and 1/D. Higher values of 1/D indicated a more diverse repertoire. There were no correlations between 1/D and absolute cell counts neither for patient nor for healthy control (HC) samples in both CD4^+^CD25^+^CD127^low^ T_reg_ and CD4^+^ T_con_ populations (Appendix A). Thus, we analyzed the CD4^+^CD25^+^CD127^low^ T_reg_ and CD4^+^ T_con_ repertoire diversity of all GVHD samples in relation to all patients without GVHD, as well as the HC group. As shown in Figure 3A, we detected a significantly lower diversity of both CD4^+^CD25^+^CD127^low^ T_reg_ and CD4^+^ T_con_ repertoires in GVHD patients than that in HCs (CD4^+^CD25^+^CD127^low^ T_reg_ cells, *p* = 0.003; CD4^+^ T_con_ cells, *p* = 0.008). Furthermore, a lower CD4^+^CD25^+^CD127^low^ T_reg_ diversity was observed in GVHD samples compared with noGVHD (*p* = 0.006), and patients without GVHD exhibited a CD4^+^CD25^+^CD127^low^ T_reg_ repertoire similar to that of healthy individuals (Figure 3A).

To further dissect the finding of reduced diversity associated with GVHD, we solely analyzed patient samples at the first occurrence of GVHD and differentiated into aGVHD and chronic or overlap GVHD (c/oGVHD). The noGVHD patient samples were matched for timing with regard to the sampling time points post DLI. For the control group, we used the average diversity of each healthy individual’s repertoires. The intergroup comparison of these subgroups revealed aGVHD to be the main driver of the observed differences in the diversity of both CD4^+^CD25^+^CD127^low^ T_reg_ and CD4^+^ T_con_ repertoires, while c/oGVHD was not statistically different from noGVHD and HCs (Figure 3B). Again, we observed healthy-like CD4^+^CD25^+^CD127^low^ T_reg_ and CD4^+^ T_con_ repertoires in noGVHD samples.

To assess diversity changes at the onset of aGVHD, we focused exclusively on the first episode of aGVHD (sampling time point median of 9 d (6–41 d) after GVHD was diagnosed) compared with the last time point prior to the onset of aGVHD (median of 66 d (5–70 d) before GVHD was diagnosed). Upon the occurrence of aGVHD, we observed a decrease in CD4^+^CD25^+^CD127^low^ T_reg_ 1/D (on average, −56.27%) and CD4^+^ T_con_ 1/D (on average, −70.29%; Figure 3C), with none of the samples meeting the interquartile range of HCs (area in blue in Figure 3C). Conversely, the repertoire diversity of matched noGVHD samples tended to increase over time in both cell populations (on average, +127.02% and +122.00%), thereby approaching healthy-like repertoire properties (Figure 3C). Comparing the change in diversity between the groups showed a statistically significant difference for the CD4^+^CD25^+^CD127^low^ T_reg_ 1/D (*p* = 0.006) but not for CD4^+^ T_con_ 1/D (Figure 3D).

Taken together, GVHD after DLI was associated with less diverse CD4^+^CD25^+^CD127^low^ T_reg_ and CD4^+^ T_con_ repertoires compared with patients without GVHD or HCs. Conversely, patients not developing GVHD after DLI showed healthy-like repertoires.

### 2.3. Acute GVHD Was Associated with Clonally Expanded CD4^+^CD25^+^CD127^low^ T_reg_ and CD4^+^ T_con_ Repertoires

To substantiate the drop in CD4^+^CD25^+^CD127^low^ T_reg_ and CD4^+^ T_con_ repertoire diversity in GVHD, which suggests the expansion of GVHD specific clonotypes, we visualized the repertoire proportion of individual clones comparing four representative samples for aGVHD, c/oGVHD, noGVHD and HCs (Figure 4A,B). To quantify these findings, we performed clonality analyses, assessing the clonal space and clonal proportion of each sample. For the clonal space, we identified the number of unique clonotypes sorted by decreasing abundancy in each sample required to occupy 25% of the whole *TRB* repertoire. Low numbers of unique clonotypes indicated a strong clonal expansion of the individual clones. For the clonal proportion analysis, we only focused on the top 20 unique clonotypes. Higher percentages of the repertoire covered by the top 20 clones indicated clonal expansion. The clonality analyses comparing all GVHD samples to noGVHD and HCs showed that fewer clones were necessary to cover 25% of the repertoire of both CD4^+^CD25^+^CD127^low^ T_reg_ and CD4^+^ T_con_ cells in GVHD patients. Similarly, GVHD patients exhibited the clonal expansion of the top 20 clones compared with patients without GVHD and HCs (Appendix A). Next, we assessed the clonality changes at the first occurrence of aGVHD regarding the top 25% of the repertoire. We observed a decrease in the number of clones, suggesting the expansion of dominant CD4^+^CD25^+^CD127^low^ T_reg_ and CD4^+^ T_con_ clones with the development of aGVHD (Figure 4C–E, Appendix A). However, the two consecutive noGVHD samples showed less focused repertoires, only meeting statistical significance for the CD4^+^CD25^+^CD127^low^ T_reg_ repertoire (*p* = 0.024; Figure 4D). This was also confirmed by the intergroup comparison (*p* = 0.015; Figure 4E). Similarly, clonal proportion analyses revealed that the top 20 clones covered an increased proportion of the CD4^+^CD25^+^CD127^low^ T_reg_ and CD4^+^ T_con_ repertoires with the onset of aGVHD, whereas matched noGVHD samples indicated healthy-like repertoire properties without clonal outgrowth (Figure 4F–H). Again, the intergroup comparison confirmed these findings with a more pronounced effect on T_reg_ cells (CD4^+^CD25^+^CD127^low^ T_re_, *p* = 0.003; CD4^+^ T_con_, *p* = 0.058; Figure 4H). To conclude, we observed clonal expansion at the occurrence of aGVHD and, on the other hand, the resemblance to healthy-like repertoires concerning patients not developing GVHD.

### 2.4. Steroid Sensitivity Was Linked to Diversification of the CD4^+^CD25^+^CD127^low^ T_reg_ and CD4^+^ T_con_ Repertoires

To gain information on the impact of steroid treatment on the CD4^+^CD25^+^CD127^low^ T_reg_ and CD4^+^ T_con_ repertoires, we plotted the repertoire diversity of respective patients over time (Appendix A). Considering the clinical response to steroid treatment, we differentiated between steroid-sensitive and steroid-refractory samples. To further quantify the observed changes in 1/D, we performed longitudinal analyses between the first sample upon steroid treatment (median of 9 d (7–40 d) after initiation of treatment) and the previous time point prior to steroid medication. In steroid-sensitive GVHD, we observed an increase in the average CD4^+^CD25^+^CD127^low^ T_reg_ repertoire diversity of 152.98% (*p* = 0.145; CD4^+^ T_con_ +900,31%, *p* = 0.277), whereas there was a drop in diversity of −58.58% in steroid-refractory GVHD (*p* = 0.448; CD4^+^ T_con_ −73,78%, *p* = 0.057; Figure 5A). Interestingly, steroid-sensitive samples presented a healthy-like repertoire diversity of both CD4^+^CD25^+^CD127^low^ T_reg_ and CD4^+^ T_con_ populations. Although the intergroup comparison revealed no significant group differences, the direct comparison of steroid-sensitive and steroid-refractory samples showed a trend towards higher CD4^+^CD25^+^CD127^low^ T_reg_ repertoire diversity (*p* = 0.054; CD4^+^ T_con_, *p* = 0.114) linked to steroid sensitivity. Clonality analyses complemented the diversity findings by showing an increased clonality in the top 25% of the repertoires as well as expanded top 20 clones in steroid-refractory samples, while the opposite was the case in the steroid-sensitive samples for both CD4^+^CD25^+^CD127^low^ T_reg_ and CD4^+^ T_con_ repertoires (Figure 5B,C).

In sum, CD4^+^CD25^+^CD127^low^ T_reg_ and CD4^+^ T_con_ repertoires in steroid-sensitive GVHD presented healthy-like repertoire properties. Conversely, we observed further clonal expansion in steroid-refractory GVHD.

### 2.5. Repertoire Properties of DLI Products and Patient Samples Prior to DLI Did Not Differ between GVHD and noGVHD

Next, we analyzed DLI products with regard to diversity, clonal space and clonal proportions. Upon the comparison of DLI products applied to patients with and without future development of GVHD, we observed no differences, neither in the CD4^+^CD25^+^CD127^low^ T_reg_ nor in the CD4^+^ T_con_ repertoire (Appendix A). Likewise, the assessment of CD4^+^CD25^+^CD127^low^ T_reg_ and CD4^+^ T_con_ repertoire diversity and clonality in patient samples taken prior to DLI did not reveal any statistically significant differences between patients with future GVHD and those without GVHD following DLI (Appendix A). Taken together, the *TRB* repertoire characteristics of the applied donor lymphocytes and patient samples prior to DLI did not predict the development of GVHD post DLI in this patient cohort.

### 2.6. Resemblance to Healthy-like Repertoires Early after DLI Coincided with a Lower Incidence of GVHD

Finally, we assessed the predictive potential of repertoire diversity properties in the first samples following DLI with regard to GVHD incidence and relapse. Therefore, we analyzed the CD4^+^CD25^+^CD127^low^ T_reg_ and the CD4^+^ T_con_ repertoire diversity (1/D) of all patient samples taken at the first time point after DLI and categorized them into expanded (<25th percentile of HC 1/D) and not expanded, i.e., healthy-like samples (1/D ≥ 25th percentile of HC 1/D). The comparison of the clinical characteristics of patients with expanded vs. healthy-like diversity revealed no significant differences between the groups (Appendix A). We then assessed the cumulative incidence of overall GVHD, aGVHD and c/oGVHD within the 36 months of study follow-up between patients with expanded and healthy-like repertoire properties (Figure 6). Within the healthy-like CD4^+^CD25^+^CD127^low^ T_reg_ repertoire, we detected a trend towards a reduced incidence of overall GVHD (HR, 0.428; 95% CI, 0.143–1.607; Figure 6A), aGHVD (HR, 0.379; 95% CI, 0.106–1.363; Figure 6B) and c/oGVHD (HR, 0.453; 95% CI, 0.098–2.089; Figure 6C). However, we did not observe such differences in the healthy-like CD4^+^ T_con_ repertoire, except for aGVHD, albeit to a lesser degree (HR, 0.655; 95% CI, 0.190–2.259; Figure 6B). To conclude, an unbiased sample analysis of the diversity properties revealed that patients with a healthy-like CD4^+^CD25^+^CD127^low^ T_reg_ repertoire early after DLI seemed to show a reduced GVHD incidence.

## 3. Discussion

While the goal of DLI treatment is to (re-)induce long-term remission via donor T-cell clones directed against recurring or persisting malignant cells, GVL often co-occurs with GVHD as result of unselective allo-reactivity [2,7]. To rule out an impact of HLA disparity on the results, only patients without HLA-mismatch were included in this trial. The rates of GVHD in our cohort were in line with published data [4]. Importantly, the occurrence of GVHD after alloHSCT did not determine the incidence of GVHD following DLI, as the groups did not differ with regard to previous GVHD. As above described, patients with GVHD after DLI had received a higher cumulative amount of donor T cells and showed a lower rate of relapse incidence but higher NRM [3]. The high rate of GVHD pointed towards the unselective allo-immune response elicited by DLI not only targeting malignant cells but also host antigens.

To date, the monitoring of GVHD following DLI depends on clinical assessment and histological confirmation. There is emerging evidence that GVHD after alloHSCT is associated with an expanded TCR repertoire, and the role as a potential biomarker has been discussed [21,22,23,24]. The identification of patients at risk of GVHD post DLI is highly relevant, as GVHD is the main driver of NRM [4]. However, in the setting of DLI, there is a lack of data evaluating the underlying immunologic dynamics at the onset of GVHD and during GVHD treatment. Employing a prospective study approach, we analyzed GVHD-dependent changes in the CD4^+^CD25^+^CD127^low^ T_reg_ and CD4^+^ T_con_ *TRB* repertoires in patients treated with DLI. Although the absolute count of immune cells did not influence the repertoire diversity (Appendix A) and a bias by divergent sequencing depth was prevented via the standardized bioinformatic analysis of all sequencing samples, we identified a markedly reduced repertoire diversity and the expansion of abundant TCR clones in patients with GVHD after DLI. Conversely, upon comparison with a cohort of healthy controls, we could show that patient samples without GVHD had healthy-like CD4^+^CD25^+^CD127^low^ T_reg_ and CD4^+^ T_con_ diversity and clonality properties. In subgroup analyses, upon the occurrence of aGVHD, we further identified a drop in diversity and the expansion of the CD4^+^CD25^+^CD127^low^ T_reg_ and CD4^+^ T_con_ top clones, while the repertoires of patients without GVHD did not show clonal expansion. These data were in line with those of previous reports showing reduced T-cell diversity in patients developing GVHD after alloHSCT based on spectratyping [23,24] and also via high-throughput sequencing [21,22], as employed in our study. However, differential sequencing analyses of T-cell subsets known to play a role in GVHD homeostasis, e.g., T_reg_ cells [9], were not included in these trials. As we previously reported diverging sequencing results in different T-cell subsets [17], we sorted the PBMCs into CD4^+^CD25^+^CD127^low^ T_reg_ and CD4^+^ T_con_ populations prior to high-throughput *TRB* sequencing. Of note, the overall repertoire dynamics of the CD4^+^CD25^+^CD127^low^ T_reg_ and CD4^+^ T_con_ populations were more pronounced in the T_reg_ compartment, especially with regard to clonality (Figure 4). Immune-cell homeostasis was shown to rely on high TCR T_reg_ diversity, including the suppression of experimentally induced aGVHD [28]. This is in line with the protection from GVHD by a healthy-like T_reg_ repertoire shown in this study. In contrast, we previously reported in a cohort of patients that expanded T_reg_ clones protected from aGVHD early after alloHSCT [20]. However, while T-cell reconstitution generally occurs within the first two months after alloHSCT, the TCR diversity remains restricted for much longer thereafter [18,21]. In addition, in the early phase after alloHSCT, inflammation is more prevalent than in the DLI setting, where immune reconstitution is more advanced. Furthermore, even if antineoplastic therapy is applied prior to DLI treatment, the degree of immune-cell repertoire reduction is less intense than that in the conditioning regimens prior to alloHSCT. Nevertheless, the observed association between GVHD following DLI and a less diverse and clonally expanded CD4^+^CD25^+^CD127^low^ T_reg_ *TRB* repertoire raises the question of why these expanded clones do not confer protection from GVHD. There are emerging approaches towards the identification of antigen specificities [29,30]; however, given the complexity of T-cell immunity, this remains a challenging and so far unsolved topic. Therefore, we can only speculate about the differences in antigen specificity between expanded T_reg_ and T_con_ clones. It might well be that the expanded T_reg_ clones in patients with GVHD following DLI exhibit a compromised stability and functional capacity. Future research should, therefore, integrate an in-depth T_reg_-cell phenotyping panel including functional markers (e.g., FOXP3 and Helios) and functional assays to assess the role of cell functionality in the protection from GVHD.

The treatment of higher-grade GVHD relies on systemic steroids as first-line therapy. Therapy might be escalated with additional immunosuppression when criteria for steroid-refractory GVHD are met [31]. Our detailed analysis of patient samples upon steroid treatment showed repertoire changes at the clonal level as a function of GVHD response. Albeit lacking statistical significance due to limited patient samples, the longitudinal analyses revealed a trend towards the diversification of the CD4^+^CD25^+^CD127^low^ T_reg_ and CD4^+^ T_con_ repertoires, approaching healthy-like properties in steroid-sensitive patients, while steroid refractoriness was associated with opposite effects upon steroid treatment (Figure 5). Evidence exists indicating that T_reg_ cells are less sensitive to glucocorticoid-induced apoptosis than CD4^+^ T cells [32]; however, we observed similar repertoire changes upon steroid treatment in both populations. Future research should investigate the role of apoptotic pathways in steroid-refractory GVHD. Furthermore, a study evaluating the TCR diversity in steroid-sensitive and steroid-refractory patients upon the diagnosis of aGVHD after alloHSCT did not show any differences [33]. However, in our cohort, we detected a trend towards higher diversity and less clonal expansion in the CD4^+^CD25^+^CD127^low^ T_reg_ repertoire in steroid-sensitive samples. The divergent results could be explained by the use of unsorted T cells for sequencing and the different setting (alloHSCT- vs. DLI-induced GVHD).

Using an unbiased sample approach comparing healthy-like to less diverse repertoires early after DLI, we identified a trend towards a reduced incidence of overall GVHD, aGVHD and c/oGVHD in patients with a healthy-like CD4^+^CD25^+^CD127^low^ T_reg_ repertoire (Figure 6A–C). Of note, this difference was observed at a median of 2.5 (0.2–15) months prior to the diagnosis of GVHD. This finding was exclusive to the T_reg_ compartment, pointing to the role of a diverse repertoire in immuno-regulation in GVHD [28]. While these results are of high interest, they warrant further testing in a large, homogenous cohort of DLI patients, before firm conclusions can be drawn.

In general, the significance of our results is limited by the small cohort size and the heterogeneity of underlying hematologic diseases. Considering the various clinical courses, including additional antineoplastic treatment concomitant to DLI therapy, we cannot exclude an impact on the observed *TRB* repertoire results. However, we ruled out that the diverging cell counts present in the patient cohort had an impact on the sequencing results. Furthermore, sequencing depth was standardized via random normalization to 20.000 reads in every sequencing sample. Despite these limitations, we were able to identify significant changes in the CD4^+^CD25^+^CD127^low^ T_reg_ and CD4^+^ T_con_ repertoires as a function of GVHD and GVHD treatment. Future studies are required to replicate these findings in a large homogenous cohort. Moreover, approaches such as data mining might even further support such validation studies.

As *TRB* repertoire analyses become more and more cost-efficient and fast, especially in a multiplex rather than unbiased RACE approach, the identification of prognostic clonotypes in a large cohort would be the key for the development of a reliable biomarker informing the diagnosis and monitoring of GVHD following DLI. Furthermore, the impact of GVHD severity and manifestation on repertoire dynamics should be investigated. Future advances might also help to identify the antigens recognized by the expanding clones, enabling the prediction of GVHD (and GVL) prior to the application of DLI. However, this remains a challenging goal considering the highly individual and thus very complex immune-cell repertoire interactions in allo-reactivity.

In conclusion, GVHD following DLI was associated with less diverse and clonally expanded CD4^+^CD25^+^CD127^low^ T_reg_ and CD4^+^ T_con_ *TRB* repertoires. Upon steroid therapy, we observed the diversification of the repertoires in steroid-sensitive GVHD patients. Moreover, a healthy-like CD4^+^CD25^+^CD127^low^ T_reg_ repertoire early after DLI was linked to a reduced incidence of GVHD during the 36 months of follow-up. While these findings warrant further testing, they hold promise to improve the diagnosis and monitoring of treatment in GVHD following DLI, thereby improving the safety of DLI.

## 4. Materials and Methods

### 4.1. Cohort and Study Design

A total of 29 patients treated with unmanipulated DLI for clinical or molecular relapse or increased host chimerism after alloHSCT were recruited between 2015 and 2017 in the department of Hematology, Hemostasis, Oncology and Stem-Cell Transplantation at Hannover Medical School. The institutional review board approved the study (#2604-2015), and written informed consent was obtained from all study participants. Patients with an HLA-mismatched donor (*n* = 5), second alloHSCT before DLI (*n* = 1), ongoing cGVHD at DLI (*n* = 1) and withdrawal from treatment (*n* = 1) were excluded from the trial. Details of study recruitment and sampling are given in Figure 1 and Appendix A. Disease risk was assessed according to ELN [26] and IPSS-R [27] criteria.

Acute GVHD (aGVHD) and (overlap) chronic GVHD (c/oGVHD) were categorized according to the EBMT-NIH-CIBMTR task force panel recommendation [31], using the Mount Sinai Acute GVHD International Consortium (MAGIC) criteria for aGVHD [34] and the National Institutes of Health (NIH) 2014 criteria for cGVHD [35]. Patients were classified as steroid sensitive or refractory upon steroid treatment based on current guidelines, respectively [31].

Moreover, 14 healthy individuals were recruited to serve as the control group for repetitive TCR repertoire sequencing over a period of six months. All healthy controls (HCs) provided their written informed consent. Smokers (*n* = 3), individuals taking immunosuppressive medication (*n* = 1) and those with cold symptoms requiring symptomatic medication on more than three concurrent study visits (*n* = 1) were excluded from the trial. Moreover, samples of healthy individuals with self-reported wounds, cold symptoms or dental procedures prior to the blood draw were excluded (Figure 1). In total, 39 samples met the inclusion criteria with a median of 4 (range of 3–6) consecutive samples per healthy individual. Details of sampling are given in Appendix A.

### 4.2. Flow Cytometry and Cell Sorting

Peripheral blood mononuclear cells (PBMCs) were extracted from patient whole-blood samples using the Ficoll gradient as described elsewhere [16] and stored at −80 °C. Thawed PBMC samples were stained at room temperature for 30 min with an antibody mix containing 7 monoclonal antibodies and 1 viability dye (Appendix A). The cell sorting of CD4^+^CD25^+^CD127^low^ T_reg_ and CD4^+^ T_con_ was performed on a BD FACSAria Fusion (BD Biosciences) cell sorter (Appendix A). Flow cytometry data were analyzed with FlowJo (version 10; TreeStar).

### 4.3. Next-Generation Sequencing and Sequence Analysis

RNA was extracted from the sorted cells; this was followed by reverse transcription using SMARTer RACE 5′-3′ PCR Kit (Clontech) using a customized protocol [20]. Next, the CDR3 region of the transcribed cDNA was amplified via RACE polymerase chain reaction (PCR) [20] with subsequent preparation for paired-end Illumina sequencing as described elsewhere [16,18]. To rule out PCR contamination, negative controls (H_2_O) were run together with the patient samples during all PCR steps.

Sequencing data analysis included the preparation of FastQ files, followed by extraction using MIXCR (version 2.1.11) with the alignment of nucleotide sequences based on the IMGT database (22.05.2018), including the correction of PCR errors. MIXCR default options were used for alignment and assembling. To avoid biased sample analysis due to diverging sequencing depths, we randomly normalized all samples to 20,000 reads using a custom bash script resulting in a standardized downsampling function. This was part of our streamline data analysis pipeline, ensuring a standardized and reproducible analysis of sequencing samples, including decompression, conversion into the FASTQ file format and random normalization (https://github.com/MHHIMMUNOLOGY/MHHTCR_ANALYSIS, accessed on 21 August 2022). For TCR repertoire analysis we used VDJtools [36] and tcR-package [37]. The inverse Simpson diversity index was used to analyze repertoire diversity [38]. Moreover, we assessed the clonal space defined by four segments, 25%, 50%, 75% and 100%, of the repertoire. For this purpose, we sorted the unique clonotypes by decreasing abundance and identified the minimal number of unique clonotypes necessary to cover each of the four defined clonal spaces. Thus, the top 25% of the clonal space of the repertoire contained the clonotypes with the highest abundance. Furthermore, the clonal proportion of each sample was investigated with regard to the most frequently occurring top 20 clones. To visualize the repertoire proportions of all unique clonotypes, we used the R (http://cran.R-project.org, accessed on 21 August 2022) *treemap* package.

### 4.4. Statistical Analysis

Statistical analyses were performed with Prism 8 (GraphPad). Normality distribution was tested with the Shapiro–Wilk test. The group comparisons of normally distributed numerical variables was conducted with unpaired two-tailed Student’s *t*-test, and non-normally distributed data were analyzed via the two-tailed, exact Mann–Whitney test. For categorical variables, Fisher’s exact test was employed. For the analysis of longitudinal changes within the same patient, paired Student’s *t*-test (normally distributed data) or the two-tailed Wilcoxon matched-pairs signed rank test (non-normally distributed data) was used. Hazard ratios for the comparison of cumulative incidence curves were calculated with the Mantel–Haenszel test. The competing risk analysis of the cumulative incidence of relapse (CIR) and non-relapse mortality (NRM), analyzing the means of cumulative incidence curves, was performed with Gray’s test [39].

## Figures and Tables

**Figure 1 ijms-23-10914-f001:**
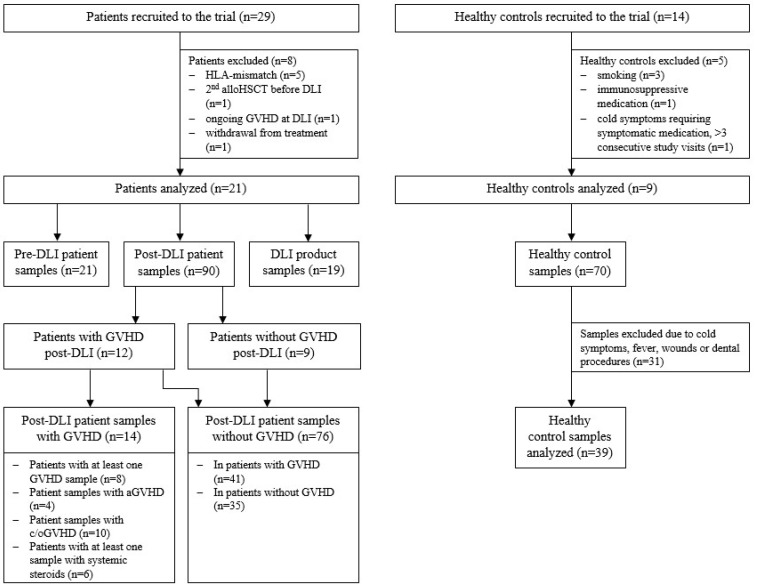
**Details of study recruitment and numbers of samples.** Abbreviations: HLA, human leukocyte antigen; alloHSCT, hematopoietic stem-cell transplantation; aGVHD, acute graft-versus-host disease; c/oGVHD, chronic/overlap graft-versus-host disease; DLI, donor lymphocyte infusion.

**Figure 2 ijms-23-10914-f002:**
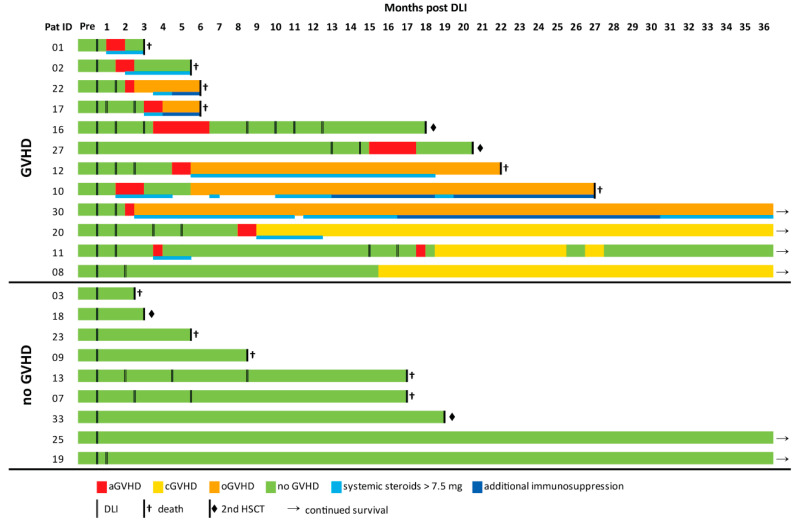
**GVHD outcome post DLI.** Lines represent individual patients’ GVHD histories during a study follow-up of 36 months; color coding indicates timing and duration of acute GVHD (red), overlap GVHD (orange), chronic GVHD (yellow), noGVHD (green), and periods of treatment with systemic steroids (>7.5 mg; light blue) and additional immunosuppression (dark blue). Application of DLI is marked by double lines. Arrows represent continued survival beyond the study follow-up, and symbols behind bold lines give details on study endpoint (♦ = second allogeneic hematopoietic stem-cell transplantation; † = death).

**Figure 3 ijms-23-10914-f003:**
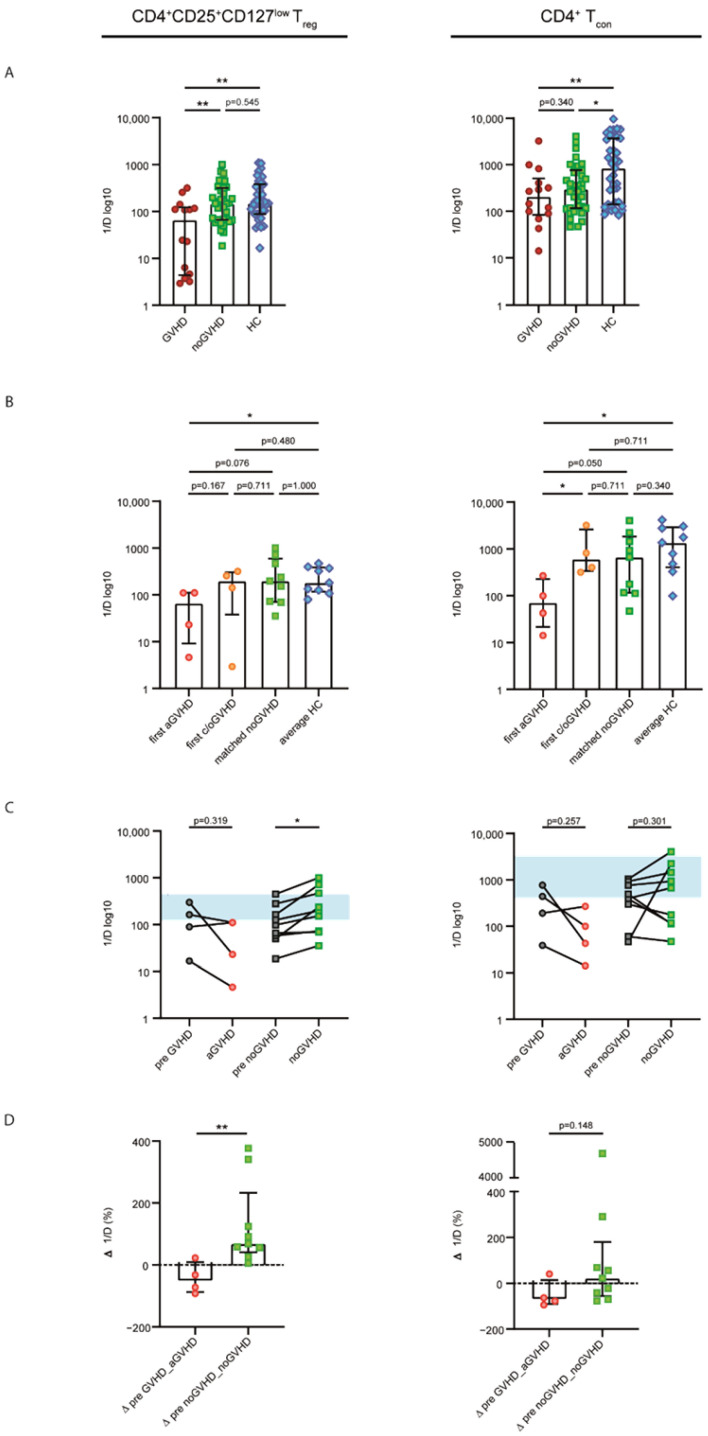
**Patients without GVHD exhibit healthy-like CD4^+^CD25^+^CD127^low^ T_reg_ and CD4^+^ T_con_ repertoires.** (**A**) Comparison of CD4^+^CD25^+^CD127^low^ T_reg_ and CD4^+^ T_con_ repertoire diversity assessed via the inverse Simpson index (1/D) in all patients’ samples with GVHD post DLI (dark-red circles; *n* = 14) or without GVHD (noGVHD; green squares; *n* = 35) compared with healthy controls (HCs; blue rhombs; *n* = 39). Higher values of the inverse Simpson index indicated a more diverse repertoire. (**B**) Comparison of samples with the first occurrence of acute (a-) GVHD (light-red circles; *n* = 4) and chronic or overlapping (c/o-) GVHD (orange circles; *n* = 4), samples without GVHD (matched for timing; *n* = 9) and average diversity in healthy controls (*n* = 9). (**C**) Analysis of CD4^+^CD25^+^CD127^low^ T_reg_ and CD4^+^ T_con_ repertoire diversity changes between the last time point prior to onset of aGVHD (pre GVHD; gray circles) and the first episode of aGVHD (light-red circles) compared with matched noGVHD samples. The blue field represents the interquartile range of the healthy controls’ average diversity. (**D**) Changes in repertoire diversity given as percentages at the onset of aGVHD compared with patients without GVHD are displayed. Longitudinal statistical analyses were performed with the Wilcoxon matched-pairs signed rank test (two-tailed) or paired Student’s *t*-test. The two-tailed Mann–Whitney test or unpaired Student’s *t*-test was used for inter-group comparisons. Black lines indicate medians, and error bars show interquartile ranges. ** *p* < 0.01; * *p* < 0.05.

**Figure 4 ijms-23-10914-f004:**
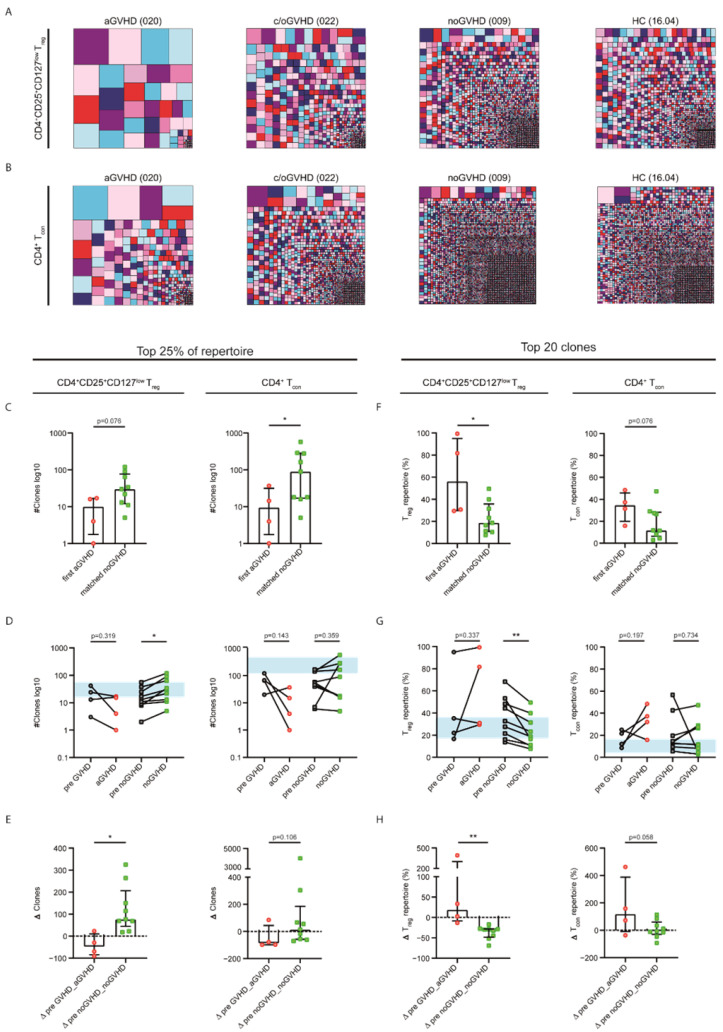
**Increased clonality in the CD4^+^CD25^+^CD127^low^ T_reg_ and CD4^+^ T_con_ compartments is associated with aGVHD**. (**A**,**B**) Tree maps visualizing the CD4^+^CD25^+^CD127^low^ T_reg_ (**A**) and the CD4^+^ T_con_ (**B**) repertoire proportions of four representative patients for aGVHD, c/oGVHD, noGVHD and healthy controls (patient IDs in parentheses). Each clone is represented by a randomly colored square with its size representing the individual clone’s proportion to the repertoire. Repetitive colors do not represent identical clones. (**C**) Numbers of unique clonotypes (y-axis) required to occupy 25% of the CD4^+^CD25^+^CD127^low^ T_reg_ and the CD4^+^ T_con_ repertoires in patients with the first occurrence of aGVHD (red circles; *n* = 4) compared with matched noGVHD (green squares; *n* = 9) samples. (**D**) Comparison of changes in numbers of unique clonotypes required to occupy the top 25% of the repertoires between the last time point prior to onset of aGVHD (pre GVHD; gray circles) and the first episode of aGVHD (light-red circles) compared with matched noGVHD samples. The blue field represents the interquartile range of the healthy controls’ average top 25% of the repertoires. (**E**) Changes in the numbers of clones given as percentages at the onset of aGVHD compared with patients without GVHD are displayed. (**F**) Clonal proportions of the CD4^+^CD25^+^CD127^low^ T_reg_ and CD4^+^ T_con_ repertoires for the top 20 clones in patients with the first occurrence of aGVHD compared with matched noGVHD samples. (**G**) Comparison of changes in the top 20 clones’ repertoire proportion between the last time point prior to onset of aGVHD (pre GVHD; gray circles) and the first episode of aGVHD (light-red circles) compared with matched noGVHD samples. The blue field represents the interquartile range of the healthy controls’ average top 20 clones. (**H**) Differences in the CD4^+^CD25^+^CD127^low^ T_reg_ and CD4^+^ T_con_ repertoires occupied by the top 20 clones given as percentages at the onset of aGVHD compared with patients without GVHD are displayed. Longitudinal statistical analyses were performed with the Wilcoxon matched-pairs signed rank test (two-tailed) or paired Student’s *t*-test. The two-tailed Mann–Whitney test or unpaired Student’s *t*-test was used for inter-group comparisons. Black lines indicate medians, and error bars show the interquartile ranges. ** *p* < 0.01; * *p* < 0.05.

**Figure 5 ijms-23-10914-f005:**
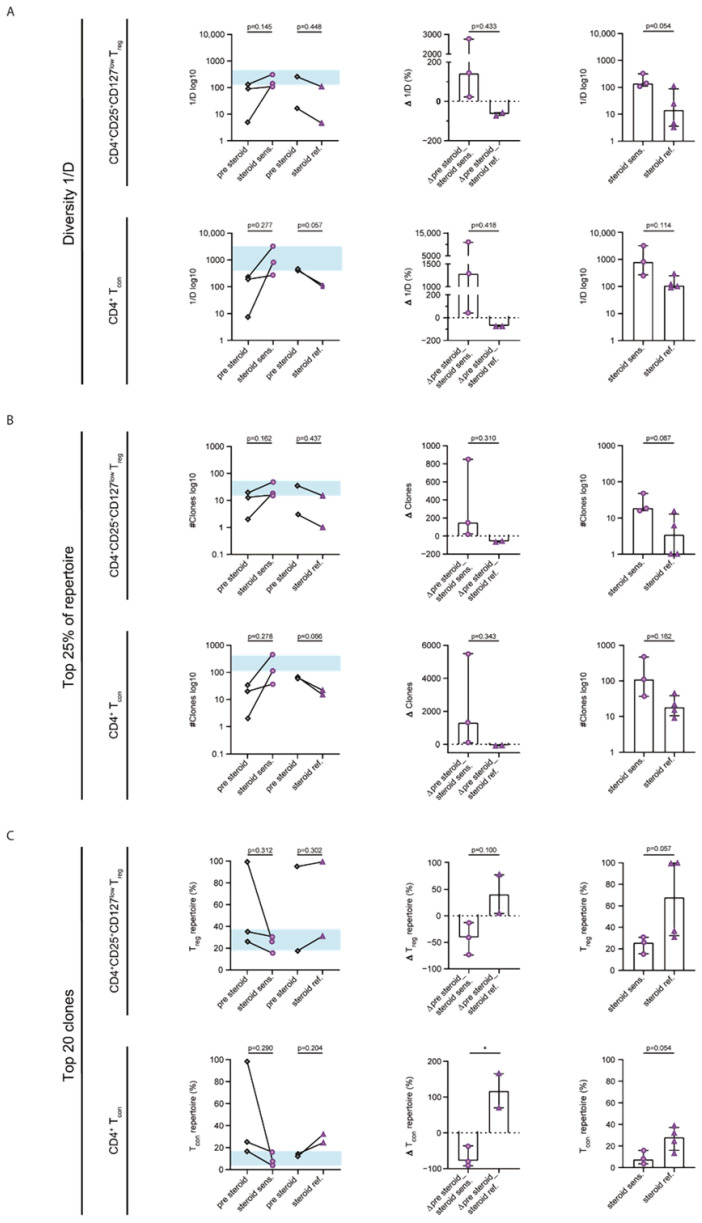
**Steroid sensitivity is linked to diversification of the CD4^+^CD25^+^CD127^low^ T_reg_ and CD4^+^ T_con_ repertoires.** (**A**) Comparison of CD4^+^ CD25^+^ CD127 ^low^ T_reg_ (upper line) and CD4^+^ T_con_ (bottom line) repertoire diversity assessed via the inverse Simpson index (1/D) in patients with GVHD receiving treatment with systemic steroids. Higher values of the inverse Simpson index indicated a more diverse repertoire. The left panel compares the last time point prior to steroid treatment (pre steroid; gray rhombs) and the first sample of steroid treatment, separated by means of steroid response in steroid sensitive (violet circles; *n* = 3) and steroid refractory (violet triangles; *n* = 2) samples. The blue field represents the interquartile range of the healthy controls’ average diversity. The center graphs compare the differences in the repertoire diversity upon steroid treatment between steroid-sensitive and steroid-refractory patients given in percentages. The right panels show an intergroup comparison of 1/D between steroid sensitive (*n* = 3) and refractory (*n* = 4) patient samples. Only for 2/4 steroid-refractory patients, there was a pre-steroid sample available, leading to reduced patient numbers for the longitudinal comparison. (**B**) Comparison of the number of unique clonotypes (y-axis) required to cover the top 25% of the CD4^+^CD25^+^CD127^low^ T_reg_ and CD4^+^ T_con_ clonal space in patient samples with either steroid-sensitive or steroid-refractory GVHD. The blue field represents the interquartile range of the healthy controls’ average top 25% of the repertoires. The panels are structured equally to those in (**A**). (**C**) The top 20 CD4^+^CD25^+^CD127^low^ T_reg_ and CD4^+^ T_con_ clones’ repertoire proportion is shown for patient samples with steroid-sensitive vs. steroid-refractory GVHD. The blue field represents the interquartile range of the healthy controls’ average top 20 clones. The panels are structured equally to those in (**A**). Longitudinal statistical analyses were performed with paired Student’s *t*-test. The two-tailed Mann–Whitney test or unpaired Student’s *t*-test was used for inter-group comparisons. Black lines indicate medians, and error bars show the interquartile ranges. * *p* < 0.05.

**Figure 6 ijms-23-10914-f006:**
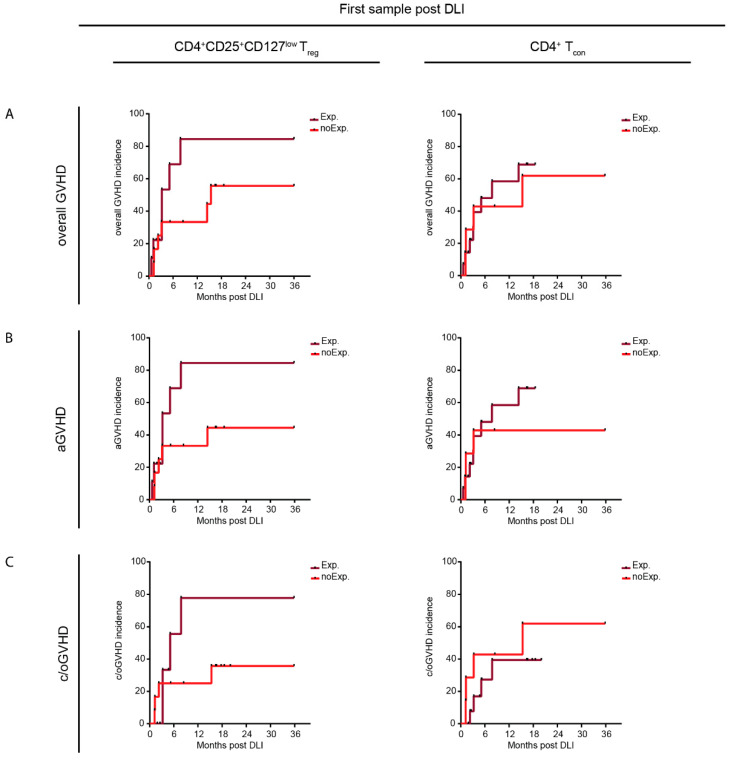
**Resemblance to healthy-like repertoires early after DLI is associated with a lower incidence of GVHD.** (**A**–**C**) The graphs display the cumulative incidence of GVHD (**A**), aGVHD (**B**) and c/oGVHD (**C**) within the 36 months of study follow-up for patients with clonal expansion (Exp.; dark-red line) and patients without clonal expansion (NoExp.; light-red line) of their CD4^+^CD25^+^CD127^low^ T_reg_ (left panel) and CD4^+^ T_con_ (right panel) repertoires at the first sampling time point post DLI (median of 14 days post DLI) compared with healthy controls (HCs). Clonal expansion was assessed via the inverse Simpson index (1/D) and compared with the 1/D of HCs. Lower values of 1/D indicated a more expanded repertoire. Patients with 1/D below the interquartile range of HCs (<25th percentile) were considered as clonally expanded samples (expanded CD4^+^CD25^+^CD127^low^ T_reg_ repertoire, *n* = 9; expanded CD4^+^ T_con_ repertoire, *n* = 14); the others were grouped into samples without clonal expansion (HC-like CD4^+^CD25^+^CD127^low^ T_reg_ repertoire, *n* = 12; HC-like CD4^+^ T_con_ repertoire, *n* = 7). (**A**) Patients with an HC-like CD4^+^CD25^+^CD127^low^ T_reg_ repertoire showed a lower GVHD incidence (HR, 0.4282 (95% CI, 0.1433–1.607) vs. 2.083 (95% CI, 0.6221–6.978)), whereas this was not seen in the HC-like CD4^+^ T_con_ repertoire (HR, 0.8156 (95% CI, 0.2542–2.616) vs. 1.226 (95% CI, 0.3822–3.933)). (**B**) Patients with an HC-like CD4^+^CD25^+^CD127^low^ T_reg_ repertoire showed a lower aGVHD incidence (HR, 0.3793 (95% CI, 0.1055–1.363) vs. 2.636 (95% CI, 0.7334–9.477)), with a similar trend in the HC-like CD4^+^ T_con_ repertoire (HR, 0.655 (95% CI, 0.1899–2.259) vs. 1.527 (95% CI, 0.4427–5.266)). (**C**) Patients with an HC-like CD4^+^CD25^+^CD127^low^ T_reg_ repertoire showed a lower c/oGVHD incidence (HR, 0.4525 (95% CI, 0.0980–2.089) vs. 2.21 (95% CI, 0.4787–10.2)), while the opposite was observed in the HC-like CD4^+^ T_con_ repertoire (HR, 2.126 (95% CI, 0.4815–9.385) vs. 0.4704 (95% CI, 0.1065–2.077)). Hazard ratios were calculated using the Mantel–Haenszel test (**A**–**C**).

**Table 1 ijms-23-10914-t001:** Cohort overview.

	All	GVHD	noGVHD	*p*-Value (GVHD vs. noGVHD)
N	21	12	9	
Sex, male	9 (43)	5 (42)	4 (44)	>0.999
Age at first DLI, years	48 (23–72)	46.5 (36–63)	51 (23–72)	0.675
Disease details: AML	15 (71)	8 (67)	7 (78)	0.659
Disease risk: adverse *	5 (24)	2 (17)	3 (33)	0.611
Donor age at alloHSCT, years	31 (22–61)	31 (22–58)	29 (22–61)	0.417
Female donor, male recipient	2 (10)	1 (8)	1 (11)	>0.999
Conditioning regimen: RIC	14 (67)	8 (67)	6 (67)	>0.999
aGVHD post alloHSCT	8 (38)	3 (25)	5 (56)	0.203
c/oGVHD post alloHSCT	2 (10)	2 (17)	0	0.486
Antineoplastic treatment pre DLI	12 (57)	7 (58)	5 (56)	>0.999
First DLI, months post alloHSCT	10.5 (4.6–75.7)	12.6 (5.6–26.3)	9.6 (4.6–75.7)	0.508
Dose of first DLI, CD3^+^/kg BW	0.5 × 10^7^ (0.1 × 10^7^–2.5 × 10^7^)	0.75 × 10^7^ (0.1 × 10^7^–2.5 × 10^7^)	0.5 × 10^7^ (0.1 × 10^7^–1 × 10^7^)	0.811
Total number of DLIs	2 (1–7)	2.5 (1–7)	1 (1–4)	0.097
Cumulative amount of applied donor cells, CD3^+^/kg BW	1 × 10^7^ (0.5 × 10^7^–32.5 × 10^7^)	4.7 × 10^7^ (0.5 × 10^7^–32.5 × 10^7^)	1 × 10^7^ (0.5 × 10^7^–5.6 × 10^7^)	0.022
Antineoplastic treatment post DLI	12 (57)	5 (42)	7 (78)	0.184
Number of relapse post DLI	12 (57)	5 (42)	7 (78)	0.184
Relapse post DLI, months	8.7 (0.2–19.3)	11.7 (6.1–19.3)	3.7 (0.2–15)	0.057
OS at 36 months post DLI, months	17.8 (1.7–36)	20.9 (2.6–36)	16.4 (1.7–36)	0.260
Patients alive at 36 months post DLI	6 (29)	4 (33)	2 (22)	0.659

Shown is the absolute number (%) for categorical variables and the median absolute value (range) for numerical variables. Fisher’s exact test (two-tailed) was used for the comparison of categorical variables; numerical variables were analyzed with unpaired Student’s *t*-test (normally distributed data) or the two-tailed Mann–Whitney test (not normally distributed data). Abbreviations: alloHSCT, allogeneic hematopoietic stem-cell transplantation; BW, body weight; DLI, donor lymphocyte infusion; aGVHD, acute graft-versus-host disease; c/oGVHD, chronic/overlap graft-versus-host disease; NRM, non-relapse mortality; RIC, reduced intensity conditioning; OS, overall survival. * Disease risk according to ELN [26] and IPSS-R [27].

## Data Availability

All data presented in this study are described in the manuscript or the Appendix A and are available upon request through the corresponding authors. All used bash and R scripts can be accessed at: https://github.com/MHHIMMUNOLOGY/MHHTCR_ANALYSIS (accessed on 21 August 2022).

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
