# Peer review of "Healthy-like CD4+ Regulatory and CD4+ Conventional T-Cell Receptor Repertoires Predict Protection from GVHD Following Donor Lymphocyte Infusion"

_ijms, 2022, doi:10.3390/ijms231810914_

Round 1

Reviewer 1 Report

Manuscript Summary:

Allogeneic hematopoietic stem cell transplantation (allo-SCT) is the most potent anti-leukemic cellular immunotherapy in patients with acute myeloid leukemia (AML). It is routinely used with curative intent in patients with an intermediate or poor-risk disease. Donor T cells, and possibly other immune cells, eliminate residual leukemia cells after prior (radio)chemotherapy, known as the is known as graft-versus-leukemia (GvL) response. However, donor alloimmune responses can also be directed against healthy tissues, which is known as graft-versus-host disease (GvHD). GvHD manifestations may cause severe complications after allogeneic hematopoietic stem cell transplantation (HSCT), involving the liver, skin, and intestine. GvHD often co-occurs with GvL, and therefore, is a major barrier to the full immunotherapeutic benefit of donor immune cells against patient leukemia because of the immunosuppression required to limit/mitigate GvHD.

This study, led by Schneider et al., follows up on the authors' past efforts on the clonal expansion of CD8+ T cells and GvL and further discusses the results of a longitudinal study evaluating the correlation between the TCR repertoire and clonal expansion of regulatory T cells (Tregs; CD4+CD25+CD127low) and conventional T cells (Tcons; CD4+) and the clinical course of GvHD in a cohort of patients receiving donor lymphocyte infusions after alloHSCT.

The manuscript is not only perfectly aligned with the aims and scope of the International Journal of Molecular Sciences but also presents several noteworthy findings that are of broad interest and importance to the field. Furthermore, the manuscript is exceptionally well-written, the results and data flow well from a logical standpoint, and acknowledges the shortcomings of the study and what appropriate steps may be pursued next to advance the field.

My only overarching question, and that too out of curiosity, is why does the clonal expansion of the Tregs not protect, at least to a partial extent, against the development of acute GvHD? Are the antigen specificities of the clonally expanded Tregs differing from the clonally expanded Tcons? Is the functional capacity or stability of these Tregs compromised in any manner?  Perhaps a Treg phenotyping panel may help elucidate these answers, especially by evaluating the levels of FOXP3, Helios, and other markers attributed to Treg function.

Minor comment: In the last paragraph in the introduction, there is a typo. 'CD4+ Tcon cells' is not formatted in a manner that is consistent with the style used across the manuscript.

Reviewer 2 Report

This paper is well-presented and the results are clear. My only concern is the small sample size, although I understand given the nature of human subject research. I wonder if a follow up study utilizing data mining or some such technique would offer more numerical support. 

Figure S1 or a version of it could be placed up front in the paper for study design clarity; S3 panel C differences in error bars are not described for future GVHD.
